# Putative Identification of New Phragmaline-Type Limonoids from the Leaves of *Swietenia macrophylla* King: A Case Study Using Mass Spectrometry-Based Molecular Networking

**DOI:** 10.3390/molecules28227603

**Published:** 2023-11-15

**Authors:** José Diogo E. Reis, Paulo Wender P. Gomes, Paulo R. da C. Sá, Sônia das G. S. R. Pamplona, Consuelo Yumiko Y. e Silva, Maria Fátima das G. F. da Silva, Anupam Bishayee, Milton Nascimento da Silva

**Affiliations:** 1Laboratory of Liquid Chromatography, Institute of Exact and Natural Sciences, Federal University of Pará, Belém 66075-110, Brazil; reisdiogo190@gmail.com (J.D.E.R.); sgsrp@ufpa.br (S.d.G.S.R.P.); yumikoyoshioka@yahoo.com.br (C.Y.Y.e.S.); 2Chemistry Post-Graduation Program, Institute of Exact and Natural Sciences, Federal University of Pará, Belém 66075-110, Brazil; 3Collaborative Mass Spectrometry Innovation Center, Skaggs School of Pharmacy and Pharmaceutical Sciences, University of California San Diego, La Jolla, CA 92093, USA; 4Skaggs School of Pharmacy and Pharmaceutical Sciences, University of California San Diego, La Jolla, CA 92093, USA; 5Federal Institute of Pará, Campus Castanhal, Castanhal 68740-970, Brazil; paulo.sa@ifpa.edu.br; 6Pharmaceutical Science Post-Graduation Program, Institute of Health Sciences, Federal University of Pará, Belém 66075-110, Brazil; 7Department of Chemistry, Federal University of São Carlos, São Carlos 13565-905, Brazil; dmfs@ufscar.br; 8College of Osteopathic Medicine, Lake Erie College of Osteopathic Medicine, Bradenton, FL 16509, USA; abishayee@lecom.edu

**Keywords:** limonoids, Meliaceae, mahogany, molecular networking, GNPS, mass spectrometry

## Abstract

*Swietenia macrophylla* King is a plant commonly known as Brazilian mahogany. The wood from its stem is highly prized for its exceptional quality, while its leaves are valued for their high content of phragmalin-type limonoids, a subclass of compounds known for their significant biological activities, including antimalarial, antitumor, antiviral, and anti-inflammatory properties. In this context, twelve isolated limonoids from *S. macrophylla* leaves were employed as standards in mass spectrometry-based molecular networking to unveil new potential mass spectrometry signatures for phragmalin-type limonoids. Consequently, ultra-performance liquid chromatography coupled with high-resolution mass spectrometry was utilized for data acquisition. Subsequently, the obtained data were analyzed using the Global Natural Products Social Molecular Networking platform based on spectral similarity. In summary, this study identified 24 new putative phragmalin-type limonoids for the first time in *S. macrophylla*. These compounds may prove valuable in guiding future drug development efforts, leveraging the already established biological activities associated with limonoids.

## 1. Introduction

*Swietenia macrophylla* King, a member of the Meliaceae family, is commonly referred to as “mahogany” in northern Brazil, and it thrives naturally in North, South, and Central America [1,2]. Its wood is renowned for its exceptional quality and high commercial value, but unfortunately, these qualities have also spurred illegal exploitation practices [2]. Conversely, various parts of the plant have been integral to traditional medicine, offering relief from an array of ailments such as malaria, anemia, diarrhea, fever, dysentery, hypertension, coughs, chest pains, intestinal parasitism, and ulcers [3].

One defining characteristic of the *Swietenia* genus is the production of phragmalin-type limonoids, which serves as a prominent biomarker for this genus in the scientific literature [4,5]. Prior research has established a significant link between these compounds and a multitude of biological activities, including hypolipidemic [6], antitumor [7], antiviral [8,9], anti-inflammatory [10,11], antifeedant [12,13], and antineuroinflammatory properties [14], among others. To date, approximately one hundred limonoids have been reported from *S. macrophylla*, including notable ones such as evodulon, gedunin, azadirone, andirobin, mexicanolide, and phragmalin, which have been documented throughout various parts of the plant [8,9,11,12,15,16,17,18,19,20,21,22,23,24,25,26,27,28,29,30,31,32,33,34,35]. Mexicanolides are primarily concentrated in the fruits and seeds, while phragmalin-type limonoids are predominantly found in the stem and leaves.

Over the past two decades, our research group has identified seventeen phragmalin-type limonoids [12,21]. However, the process of isolating and determining the structure of these limonoids remains challenging, often taking a significant amount of time, ranging from months to years [4,5]. To address these limitations, liquid chromatography-mass spectrometry (LC-MS) has emerged as the leading approach for the putative annotation of molecules within complex matrices, utilizing a straightforward LC-MS/MS analysis [36] with minimal sample quantities. This approach is robust, reliable, and efficient [37]. Additionally, advanced bioinformatic methods such as ion identity molecular networking (IIMN) [38] have been developed to enhance data interpretation and metabolite annotation. IIMN generates molecular clusters through spectral similarity comparisons, facilitating the annotation of different ions while cross-referencing them against reference spectra from open MS libraries available on the Global Natural Products Social Molecular Networking (GNPS) platform [39,40].

Hence, we employed ultra-high-performance liquid chromatography coupled with high-resolution mass spectrometry for data acquisition. Subsequently, we utilized the GNPS platform to generate molecular networks. Additionally, we employed twelve isolated phragmalin-type limonoids as standards (level 1 identification by NMR data) to uncover new putative limonoids (level 3) from the leaves of *S. macrophylla* (Figure 1).

## 2. Results

### 2.1. Standards and Their Structural Characteristics

The dichloromethane extract (DCMEt) and 12 standards were subjected to UHPLC-MS/MS in the positive ionization according to previous methods of limonoid ionization in the plants from the *Swietenia* genus [12,14,21]. Figure 2 shows total ion chromatograms (TIC) for the DCMEt extract and the standards in positive ionization mode (ESI^+^). Appendix A shows the standards (**1**–**12**) and the putatively annotated compounds (**13**–**36**), and most of them are described for the first time in the leaves of the *S. macrophylla*. We highlight that for those putatively annotated compounds, isolation and NMR experiments are still needed and are crucial for advancing our understanding of the three-dimensional structure of limonoids. In addition, we created a GNPS library for the limonoids and it can be found at the following link https://gnps.ucsd.edu/ProteoSAFe/result.jsp?task=5e75cee226b74a7a9b4f140a67bae579&view=group_new_annotations_db, accessed on 15 August 2023.

Twelve isolated phragmalin-type limonoids (**1**–**12**) were used in the molecular networks as standards (level 1 identification), and then, new putative limonoids from *S. macrophylla* leaves were annotated at level 3 [41] based on MS/MS similarity between standards and unknown compounds. Hence, two strategies were applied: (1) Standards—assignment of characteristic neutral losses to the functional groups, i.e., furan, benzoate, acetate, tiglate, hydroxyl, and orthoester, as well as diagnostic ions (MS^2^) from the carbon skeleton; (2) Unknown compounds—putative limonoids were investigated by the mass difference and edges annotation, as such, they shared neutral losses with standards. Moreover, the LOTUS database (https://lotus.naturalproducts.net/, accessed on 15 August 2023) [42] was used to search for the term “limonoid” and it returned 2968 structures. These structures were useful in the annotation process, i.e., in the attribution of novelty to the putative annotation described in this study. In addition, common functional groups present in the different types of limonoids retrieved from LOTUS such as furan, benzoate, tiglate, acetate, and so on were used to create an in-house database manually, containing 270 combinations of putative phragmalin limonoids (Appendix A) [11,12,13,14,21,26,33,43,44].

The carbon skeleton of the standards showed at least a tiglate or benzoate group linked at C-3, an 8,9,30-orthoester unit (O_3_-R, R = methyl, isopropyl, tigloyl or 2-methylbutyl), acetate at C-6 or C-12, and hydroxyls at C-1 and C-2 (Figure 3). Furthermore, the main fragmentation pathways are shown in Figure 3A–F, the neutral losses of H_2_O [(M + H) − 18]^+^, that occur most frequently from the parental ion, suggest the presence of hydroxyl groups (Figure 3A), and neutral losses of C_2_H_4_O_2_ [(M + H) − 60]^+^ by remote hydrogen or McLafferty rearrangement [45,46,47], or C_2_H_2_O and H_2_O [(M + H) − 42 − 18]^+^ via cleavage of ester bonds [45,46], followed by a neutral loss of H_2_O, suggest the presence of acetate groups (Figure 3B). Other losses of C_2_H_4_O_2_ can occur from the carbomethoxy group at C-6 (Figure 3C), which may explain the additional losses of 60.02 Da, even after the acetate group leaves or in the absence of it, that occurs in some compounds annotated in this work. The tiglate and benzoate groups (Figure 3D) can be characterized by neutral losses of C_5_H_6_O (82.04 Da) and C_7_H_6_O_2_ (122.03 Da), respectively. The presence of a typical D-ring substituted with 17-furan in most standards, except limonoid **1** which has a modified furan group, it is characterized by a neutral loss of C_5_H_4_O_2_ (96.02 Da) using the retro Diels–Alder (RDA) reaction (Figure 3E), and an elimination of a carbonyl fragment added to the elimination of a conjugated alkene. The reaction highlighted in Figure 3F suggests the presence of the methyl group in 8,9,30-orthoester units, such as in limonoids **2** and **3**. Otherwise, the neutral losses of C_4_H_8_O_2_ [(M + H) − 88]^+^ or C_4_H_6_O, and H_2_O [(M + H) − 70 − 18]^+^ suggests the presence of the isopropyl group, as shown in limonoids **4**–**7**. In addition, the neutral loss of C_5_H_8_O_2_ [(M + H) − 100]^+^ suggests the presence of the tigloyl group in limonoid **8**. Lastly, the 2-methylbutyl group can be confirmed by the neutral losses of C_5_H_10_O_2_ [(M + H) − 102]^+^ or C_5_H_8_O, and H_2_O [(M + H) − 84 − 18]^+^ as observed for limonoids **1** and **9**–**12** [45,46,47].

### 2.2. Molecular Networking and Putative Annotation

The molecular network consisted of 218 unique nodes of *m*/*z* and a retention time in which we putatively annotated 24 new limonoids. For example, the precursor ion [M + H]^+^ of *m*/*z* 713 (**30**) showed high spectral similarity to the *m*/*z* 699 (**2**) standard. However, a single difference of +14 Da (CH_2_) was observed in compound **30**. Manual curation of the MS/MS spectra indicated a new limonoid with modification at the orthoester once the fragment of the orthoester group from standard (**2**) was not found in compound **30** (Figure 4A). Another example is limonoid **34** [M + H]^+^ of *m*/*z* 761.2812, which showed similar fragmentation to standard **5** (cosine > 0.67). The main fragments were *m*/*z* 743, 661, 647, 643, 625, 565, 547, 539, 529, 521, 497, 479, 461, and 451. Limonoid **34** had a tigloyl group linked to a 8,9,30-orthoester unit (Figure 4B). This was suggested by the fragment of *m*/*z* 643 by neutral losses of C_5_H_8_O_2_ and H_2_O from the ions of *m*/*z* 743 and 661, which derive from the parental ion (*m*/*z* 761). The most intense ion (*m*/*z* 743) observed in the MS^2^ spectrum of **34** corresponded to the first loss of H_2_O, which in this case occurred at C-1. From the ion of *m*/*z* 661 to 539 occurs the loss of C_7_H_6_O_2_ [(M + H) − C_5_H_8_O_2_ − C_7_H_6_O_2_]^+^, suggesting a benzoate group linked at C-3. The presence of the acetate group at C-6 was suggested by the fragment of *m*/*z* 479, which could occur via two pathways: (1) direct loss of C_2_H_4_O_2_ from the ion of *m*/*z* 539, or (2) sequential losses of C_2_H_2_O (from *m*/*z* 539 to 497) and H_2_O. The furan group could be explained by an ion of *m*/*z* 565 from 661. Thus, this compound was putatively annotated as 3-detigloyl-3-benzoyl-6-acetoxyl-12-deacetoxyl-8,9,30-*ortho*-tigloylate-swietemacrophine. The other standards had the main fragmentation pathways discussed previously, and an overview of these compounds and their structures in the molecular network is shown in Figure 4C.

In particular, limonoid **31** [M + H]^+^ of *m*/*z* 741.3121 (Figure 5A2) was annotated as an isomer of standard **11** (Figure 4C), based on their identical MS^2^ spectrum. It was observed that unlike standard **11**, which has an acetate group at C-12, the acetate group in limonoid **31** was linked at C-6. This compound was thus characterized as 2-dehydroxyl-6-acetoxyl-12-hydroxyswietephragmin E, with product ions of *m*/*z* 723, 657, 639, 627, 621, 579, 539, 521, 479, and 461. The precursor ion (*m*/*z* 741) underwent losses of H_2_O, C_5_H_8_O, and C_5_H_10_O_2_ to generate fragments of *m*/*z* 723, 657, and 639, respectively. Additionally, the product ion of *m*/*z* 639 was generated after a further neutral loss of H_2_O from the product ion of *m*/*z* 657. The loss of H_2_O could occur at either C-1 or C-12, while the neutral losses of C_5_H_10_O_2_ [(M + H) − 102]^+^, C_5_H_8_O, and H_2_O [(M + H) − 84 − 18]^+^ suggested the presence of the 2-methylbutyl group linked to the O3-R. The loss of C_5_H_4_O_2_ (96.02 Da) from the fragment of *m*/*z* 723 generated the fragment of *m*/*z* 627, while the loss of H_2_O from *m*/*z* 639 to 621 was followed by a neutral loss of C_5_H_8_O_2_ (100.05 Da) to generate *m*/*z* 521. These fragmentation pathways confirmed the presence of the furan group (C_5_H_4_O_2_) at C-17. The ion of *m*/*z* 539 was generated from *m*/*z* 639 via the neutral losses of the tiglate group, followed by losses of acetate (*m*/*z* 479), and H_2_O (*m*/*z* 461), respectively.

In contrast to the previously described limonoids, the analysis of the fragmentation patterns of limonoids **17**, **19**, **23**, and **29** suggests possible unsaturation between positions C5-C6 or C11-C12 (Figure 5), which has not been reported in the literature. Limonoids **17** (2-dehydroxyl-12-acetoxyswietephragmin I) and **23** (2-dehydroxyl-6-acetoxyswietephragmin I) were characterized as compatible isomers with the molecular formula C_36_H_40_O_13_, which can be distinguished by the positions of the unsaturation and of the acetate group. Assuming that **17** and **23** can be derived from standards **2** and **3** via the loss of H_2_O (*m*/*z* 699.2 to 681.2), the unsaturation could be located either between C5-C6 or C11-C12, and therefore the acetate group can be bound at C-6 or C-12. Several fragments are common to **3**, including *m*/*z* 639, 621, 603, 579, 539, 521, 503, 497, 493, 479, and 461. The ion of *m*/*z* 639 corresponds to a loss of C_2_H_2_O [(M + H) − C_2_H_2_O]^+^, followed by a loss of H_2_O (*m*/*z* 621) in the orthoester group. In addition, the ion of *m*/*z* 621 indicates loss of the orthoester group (60.02 Da) from the precursor ion, which explains the 8,9,30-ortho-methylate unit in the limonoids. The loss of the tiglate group was confirmed by ions of *m*/*z* 539, 521, and 479, and of the acetate group by the ion of *m*/*z* 579. From the ion of *m*/*z* 521 to 493 there was a loss of CO, followed by a loss of CH_4_O (*m*/*z* 461). The fragments of *m*/*z* 603 and 503 were explained by further losses of H_2_O in **17** and **23**, and the ion of *m*/*z* 497 corresponds to the loss of C_2_H_2_O in **17**. Furthermore, the position of the unsaturation and the acetate group was deduced from the analysis of the fragmentation of standards and high cosine similarity between two MS/MS matching spectra in the molecular network.

All the limonoids discussed thus far have contained an intact furan ring. However, it is noteworthy that limonoids of the phragmalin-type with a modified furanic ring are a minority. In fact, our group was the first to publish a limonoid of this type in *S. macrophylla* [12] using NMR data (**1**). Now, we are presenting six limonoids with modified furan ring type 20,21,22,23-diepoxyfuran (**13**–**15**, **18**, **20**, and **24**). In these limonoids, similar variations of substituent groups are observed. For instance, a tiglate or benzoate group at C-3, and methyl, isopropyl, or 2-methylbutyl groups attached to the orthoester group, as well as hydroxyl and acetate groups, which can interchange their positions in the limonoid structure. The lack of the typical furan ring may be attributed to the absence of neutral losses of 96.02 Da, which are distinctive features of most of the limonoids described in this study.

## 3. Discussion

Limonoids have been identified as a new source of biological activity in recent investigations with Swietenia species, including anti-inflammatory [48], anti-diabetic [49,50], anti-inflammatory [48], and anticancer [51] properties. 6-*O*-acetylswietephragmin E (11) has been shown to be a strong anticancer agent in the therapy of colorectal cancer, which is the second most frequent cancer in women and the third most common in men globally [52]. This compound is one of the limonoids with potential pharmaceutical applications. Therefore, it is necessary to create a productive workflow for finding novel limonoids. In this investigation, a novel approach utilizing UPLC-HRMS/MS was applied to the DCMEt extracts of *S. macrophylla* leaves in order to identify twenty-four potential new limonoids. Molecular networks were also created using feature-based molecular networking [39,40,53,54,55,56]. This is the first study to report phragmalin-type limonoid propagation (level 3 identification) by MS-based molecular networking from standards of limonoids at level 1 of identification. Level 1 corresponds to an unequivocal identification by isolation and structure elucidation of the metabolite through nuclear magnetic resonance spectroscopy (NMR). On the other hand, level 2 is attributed to a probable structure based on a reference spectrum, which are usually spectra available on MS Tandem databases [41].

Overall, all phragmalin-type limonoids that were annotated in this study have a functional group such as hydroxyl, acetate, furan, tiglate, benzoate, and orthoester linked to the carbon skeleton illustrated in Figure 1. The presence of hydroxyl groups explains water losses at C-1, C-6 or C-12. This fragmentation may occur through two pathways: via (1) cleavage of ester bond at C-6 or C-12 [57,58,59], and (2) cleavage at the 8,9,30-orthoester unit. The neutral loss of acetate can also come from the carbomethoxy group at C-6 or at C-12. Usually, phragmalin-type limonoids have a common furan group at C-17, and we observed a neutral loss of 96.02 Da, potentially associated to that group; its fragmentation pathway can be explained by retro Diels–Alder (RDA) in the D-ring, except for the compounds **1**, **13**–**15**, **18**, **20**, and **24**, due to their modified furan group. The presence of the tiglate group at C-3 could be confirmed by two pathways: a neutral loss of C_5_H_6_O or C_5_H_8_O_2_. The specific left group is determined by whether one hydroxyl group is present or absent at C-2. Otherwise, a neutral loss of C_7_H_6_O_2_ was present in all limonoids with benzoate groups. Lastly, orthoester groups such as 8,9,30-ortho-methylate, 8,9,30-ortho-ethylate, 8,9,30-ortho-isobutylate, 8,9,30-ortho-2-methylbutenoate, and 8,9,30-ortho-tigloylate were characterized via losses of C_2_H_4_O_2_, C_3_H_6_O_2_, C_4_H_8_O_2_, C_5_H_10_O_2_, and C_5_H_8_O_2_, respectively (Appendix A). Finally, typical losses of CO_2_ and CO were noticed.

To date, phragmalin limonoids containing the 8,9,30-orthoester group were reported once in *Swietenia mahogany* [43], *Xylocarpus granatum* [60,61,62], and *Chukrasia tabularis* [63]. In this context, based on the standards and their fragmentation patterns, here we described a couple of them, making *S. macrophylla* the fourth species known to produce these compounds. Notably, in this study, four phragmalin orthoester limonoids (**21, 22, 31**, and **36**) are described, potentially for the first time. Furthermore, eleven other limonoids (**17, 19**–**23, 27, 29, 31, 33**, and **36**) did not exhibit hydroxyl or acetate groups at C-2, which is a common modification recently reported in an isolated limonoid from the roots of *S. macrophylla* [33]. Along with hydroxyl modification at C-2, three limonoids with hydroxylation at C-6 (**20**, **27**, and **33**) were also putatively described for the first time, and so far, modification at C-6 of limonoids has been reported only in *S. mahogany* [43]. Additionally, we observed limonoids with a 20,21,22,23-diepoxyfuran ring, which is uncommon among Meliaceae species [4]. So far, only one phragmalin-limonoid with that ring has been described in *S. macrophylla* [12]. In this context, herein, we are revealing six new phragmalins with the 20,21,22,23-diepoxyfuran ring (compounds **13**–**15**, **18**, **20**, and **24**).

## 4. Materials and Methods

### 4.1. Chemicals

Ultrapure water was obtained from a Merck Millipore Milli-Q^®^ Direct 5 UV system (Darmstadt, HE, Germany). Acetonitrile (CH_3_CN) and methanol (LC-MS grade) were purchased from Merck (Darmstadt, HE, Germany). Formic acid (LC-MS grade) from SK Chemicals (Seongnam, GG, Republic of Korea). Dichloromethane (ACS grade) was purchased from Tedia (Fairfield, OH, USA), and ethyl alcohol 99.5% (*v*/*v*) was purchased from Soltech (Diadema, SP, Brazil). Twelve isolated limonoids from *S. macrophylla* leaves were used as standards (identification at level 1). These compounds were isolated by the efforts of our group in the last two decades [12,21].

### 4.2. Plant Collection

The leaves of *S. macrophylla* were collected at José da Silveira Neto University City (1°28′24″ S; 48°27′11″ W), Belém, Pará, Brazil. Dr. Orlando Shigueo Ohashi identified the species, which was then deposited in the Federal Rural University of the Amazon’s Herbarium (voucher specimen: 1330a). The National System for Managing Genetic Heritage and Associated Traditional Knowledge (SISGEN) issued authorization for us to access the Brazilian genetic heritage with the permission code A678D8C.

### 4.3. Extraction

The leaves of the *S. macrophylla* were washed with water and sodium hypochlorite 2.5% (*v*/*v*), and sprayed with ethyl alcohol 70% (*v*/*v*), respectively. The leaves were dried in an air circulation oven (Quimis, Diadema, SP, Brazil) at 45 °C until constant in weight, and then the dry leaves were crushed using grail and pestle, until a granulometry of 60–100 µm. After that, a total of 25 g of crushed material was extracted with 500 mL of dichloromethane, and the extractive process (2 batches of 30 min at 45 °C) was performed using the ultrasonic bath model Branson 2510MT (Marshall Scientific, Hampton, NH, USA). Lastly, the resultant extract solutions were dried under vacuum in a rotary evaporator (Buchi, Flawil, Switzerland), combined, and dried at 45 °C until a constant weight, resulting in 6.1 g (24.4% yield).

### 4.4. LC-MS/MS Analysis

The sample pretreatment involved the use of a C18 Solid Phase Extraction (SPE) cartridge (50 mg) (Torrance, CA, USA), which had been conditioned beforehand with 1 mL each of CH_3_CN, and water (H_2_O). A total of 10 mg of dried extract suspended in 1 mL of H_2_O/CH_3_CN (20:80, *v*/*v*) was filtered in the C18 cartridge, yielding 3.2 mg. The resulting solution was suspended in 3.2 mL of H_2_O/CH_3_CN (20:80, *v*/*v*), and then it was filtered through a 0.22 µm hydrophilic syringe filter (Merck Millipore, Darmstadt, HE, Germany). The resulting solution (1 mg·mL^−1^) was injected into the Ultra Performance Liquid Chromatography (UPLC) system coupled to the Q-Tof Xevo G2-S mass spectrometer (Waters Corp., Milford, MA, USA) with an electrospray ionization source (ESI). The reference mass used was Leucine-enkephalin (Waters Corp., Milford, MA, USA). The chromatographic experiment was carried out using a BEH C18 column (50 × 2.1 mm, 1.7 µm, Waters Corp.) at a column temperature of 40 °C and a flow rate of 300 µL·min^−1^. H_2_O (solvent A) and CH_3_CN (solvent B), acidified with 0.1% formic acid, were used as mobile phases. A linear gradient from 35 to 55% CH_3_CN (*v*/*v*) was applied, with a total time of 30 min. In the same batch, a pool containing 12 standards was analyzed and their retention time and MS information were manually mapped.

Mass spectra acquisition in positive mode ionization (50–1200 Da) was performed using the MassLynx 4.1 software (Waters Corp., Milford, MA, USA). The scan time was set to 0.5 s and the instrument settings were optimized: source temperature at 120 °C; cone gas (N_2_) flow of 50 L/h; and desolvation gas (N_2_) flow of 600 L/h at 150 °C. The capillary and cone voltages were set at 3.0 kV and 40 V, respectively. A data-dependent analysis (DDA) method was performed (centroid format), and the five most intense ions (Top5 experiment, 0.5 s, and charge states of +1 and +2) were chosen for MS/MS acquisition with normalized collision energy (NCE) from 10 to 50 eV. The tolerance and peak extract window of the parent ion were set to ±0.2 and 2 Da, respectively, with a deisotope tolerance of ±3 Da and an extraction tolerance of deisotope 6 Da.

### 4.5. Data Processing (MZmine 3)

The LC-MS/MS data obtained in RAW format from Waters Corp., Milford, MA, USA were converted into the mzXML format using MSConvert 3.0.2 [54]. The MS features were retrieved using MZmine 3.4 [64]. For mass detection, signal noises of 1.0 × 10^4^ and 1.0 × 10^2^ were used for MS^1^ and MS^2^ levels, respectively. The ADAP chromatogram builder was utilized for chromatogram construction, with a minimum group size of scans of 3, minimum intensity of the group set to 1.0 × 10^4^, and the highest to 3.0 × 10^4^ with an *m*/*z* tolerance of 10 ppm. The ADAP resolver was employed for chromatographic deconvolution, and intensity window S/N was used as an S/N estimator, with a signal-to-noise ratio of 10. A minimum feature height of 1.0 × 10^4^, a coefficient of peak area of 1.70, a peak duration from 0.05 to 2.0 min, and an RT wavelet range used to build a matrix of coefficients from 0.05 to 0.10 min were used. The isotope peak grouper module was applied to detect the isotopes with an *m*/*z* and RT tolerance of 10.0 ppm and 0.2 min, respectively, with charge 1 as the standard. The resulting peak list was filtered to remove duplicate features and aligners. The weight of *m*/*z* and RT was set to 75:25, respectively. The resulting peak list was then filtered to remove duplicated features as well as from the blanks, and only features with isotope patterns and MS^2^ spectra were kept. The filtered peak list containing 218 features was exported as .mgf and .CSV files containing feature information, such as retention time, peak area, peak intensity, MS^1^, and MS^2^.

### 4.6. Ion Identity Molecular Networking (IIMN)

A table (.CSV) containing the metabolite information, such as retention time, precursor mass, peak intensity, and the peak list (.MGF) with MS/MS data, was used to build molecular networks with ion identity in the GNPS platform (http://gnps.ucsd.edu, accessed on 19 April 2023). A tolerance of 0.02 Da was applied to search for precursor ions (MS^1^) and product ions (MS^2^). The minimum cosine score was set to 0.60 and the minimum matched fragment ions to 3. The maximum size of the molecular family was tuned at 100, and the maximum of neighbor nodes was set to 10. The same settings were applied to the library searches, and the generated molecular networks (https://gnps.ucsd.edu/ProteoSAFe/status.jsp?task=8b7d836ad2694de0b932cf65db8c3fae, accessed on 19 April 2023) were visualized in the Cytoscape 3.8.2 software [53].

## 5. Conclusions

The ability of molecular networks to rapidly detect and annotate new compounds, such as those reported in this study, using mass spectrometry based on the fragmentation patterns of previously isolated and identified compounds, was well-established. Through this approach, we were able to analyze the abundance and biosynthetic variety of limonoids. These limonoids are significantly more complex than other triterpene derivatives due to their extensive changes in the carbon skeleton. This was made possible by combining molecular networks and LC-MS/MS data. To address this challenge, the chemical information obtained from the MS/MS data of the standards provided valuable insights into the main neutral losses and functional groups involved in structural diversification. This information helped us to establish the fragmentation pathway of the new limonoids. Consequently, we could demonstrate that all 24 phragmalin-type limonoids possess an 8,9,30-orthoester unit, and we could identify the attached substituents. We also unequivocally determined the substituents on the C-3 positions, either benzoate or tiglate. This study generated a spectral library for phragmalin-type limonoids, which was previously unavailable. This library can be used to characterize similar skeletons (tetranortriterpenoids) with greater precision and provide highly reliable confirmatory measurements for field research. Moreover, the workflow used in this study can optimize data processing time in studies involving complex natural products, eliminating the need for exhaustive isolation procedures and expediting the discovery of biologically active natural compounds. However, it is essential to acknowledge certain limitations. While our aim is consistent and rapid chemical characterization using tandem mass spectrometry data, we can only assign putative annotations, which represent the planar structure. Therefore, considering these limitations, we emphasize the necessity of NMR experiments on the compounds annotated here. Such experiments remain crucial for identifying the three-dimensional structure of these molecules.

## Figures and Tables

**Figure 1 molecules-28-07603-f001:**
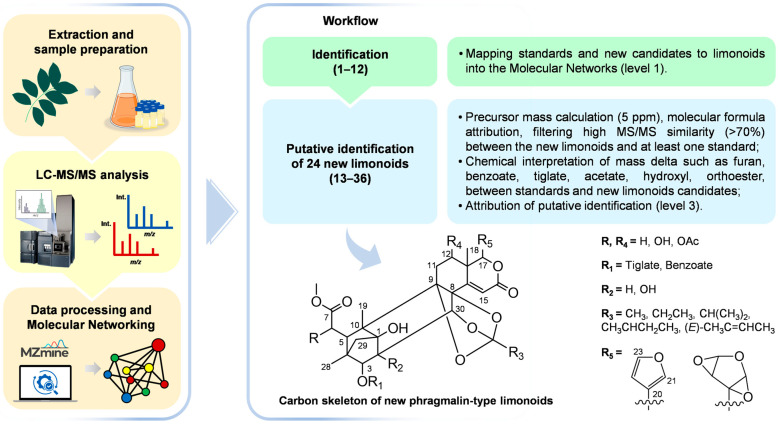
Workflow to putative identification of new phragmaline-type limonoids from the leaves of *Swietenia macrophylla* King.

**Figure 2 molecules-28-07603-f002:**
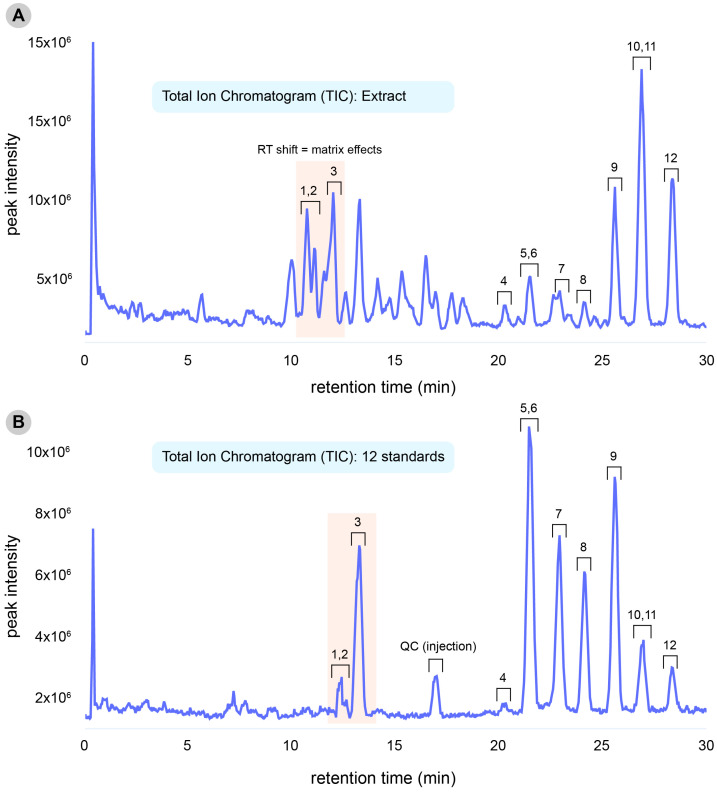
Total Ion Chromatograms (TIC) from *Swietenia macrophylla* King leaves for DCMEt extract (**A**) and 12 standards (**B**) were obtained using UHPLC-HRMS in positive ionization mode (ESI^+^). The RT shift to compounds **1**–**3** is explained by the matrix effect in the extract caused by many compounds in the same range.

**Figure 3 molecules-28-07603-f003:**
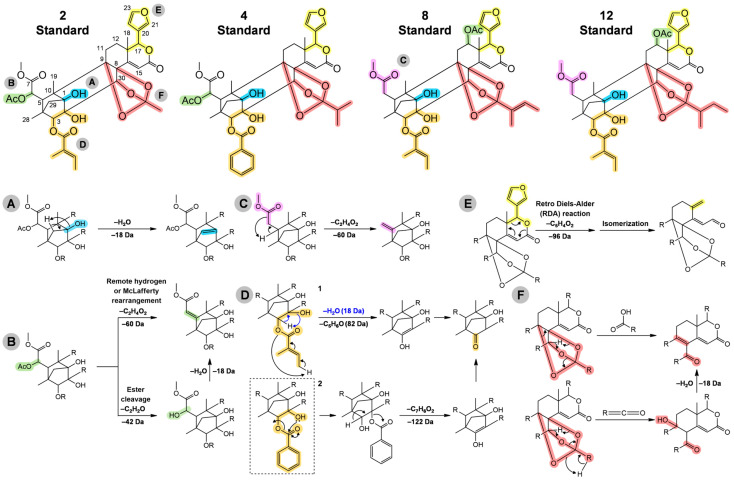
General fragmentation pathways proposed for the standards (isolated limonoids). Main functional groups and reactions are highlighted: (**A**) hydroxy in blue shadow; (**B**) acetate in green shadow; (**C**) carbomethoxy in pink shadow; and (**D**) tiglate or benzoate in orange shadow. The neutral loss of H_2_O described in mechanism D1 could also occur via mechanism A, followed by loss of tiglate; (**E**) furan in yellow shadow; and (**F**) orthoester in red shadow. Loss of R-CO and R-COOH are characteristic for the 8,9,30-orthoester unit (R could be a methyl, isopropyl, tigloyl, or 2-methylbutyl).

**Figure 4 molecules-28-07603-f004:**
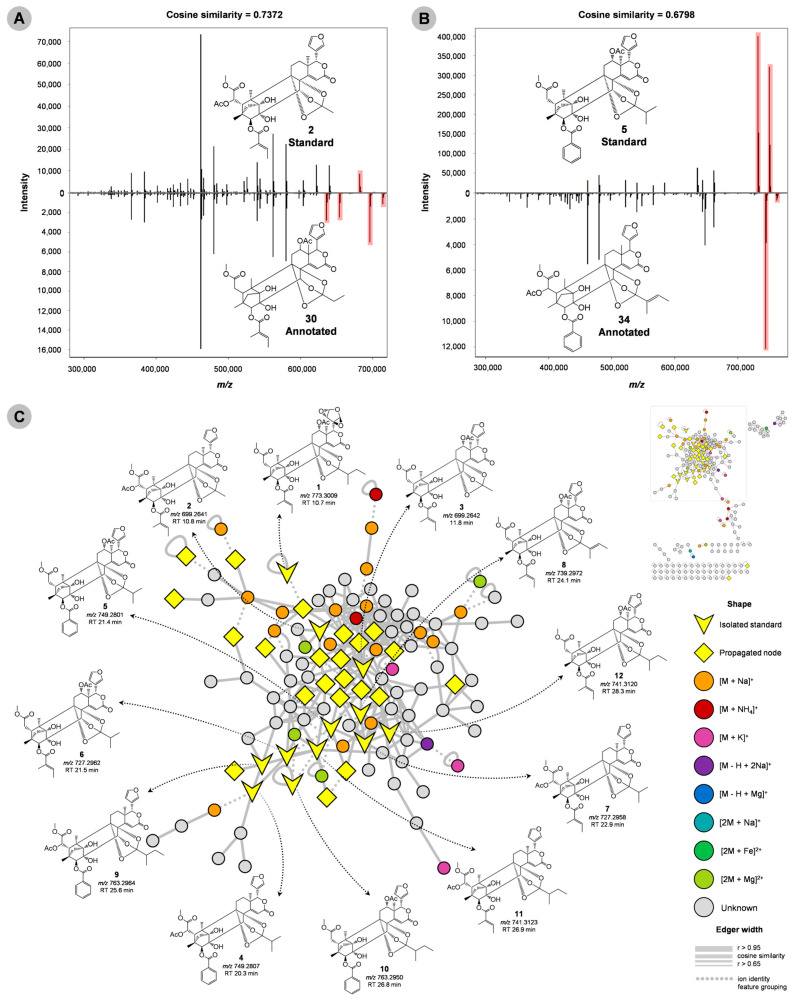
Matches between reference MS/MS spectra of standards and experimental MS/MS spectra of annotated limonoids, considering the mass shift/difference in red shadow (**A**,**B**). Molecular network of the main cluster (**C**). The V-shaped node corresponds to isolated limonoids and their respective structure. The text below the structures indicates the parent ion and RT of the isolated compounds. The thickness limit (the line that connects two nodes) represents a spectral similarity (thicker, more similar MS Spectrum 0.65 < r > 0.95) and the dotted edge shows ion identity feature grouping.

**Figure 5 molecules-28-07603-f005:**
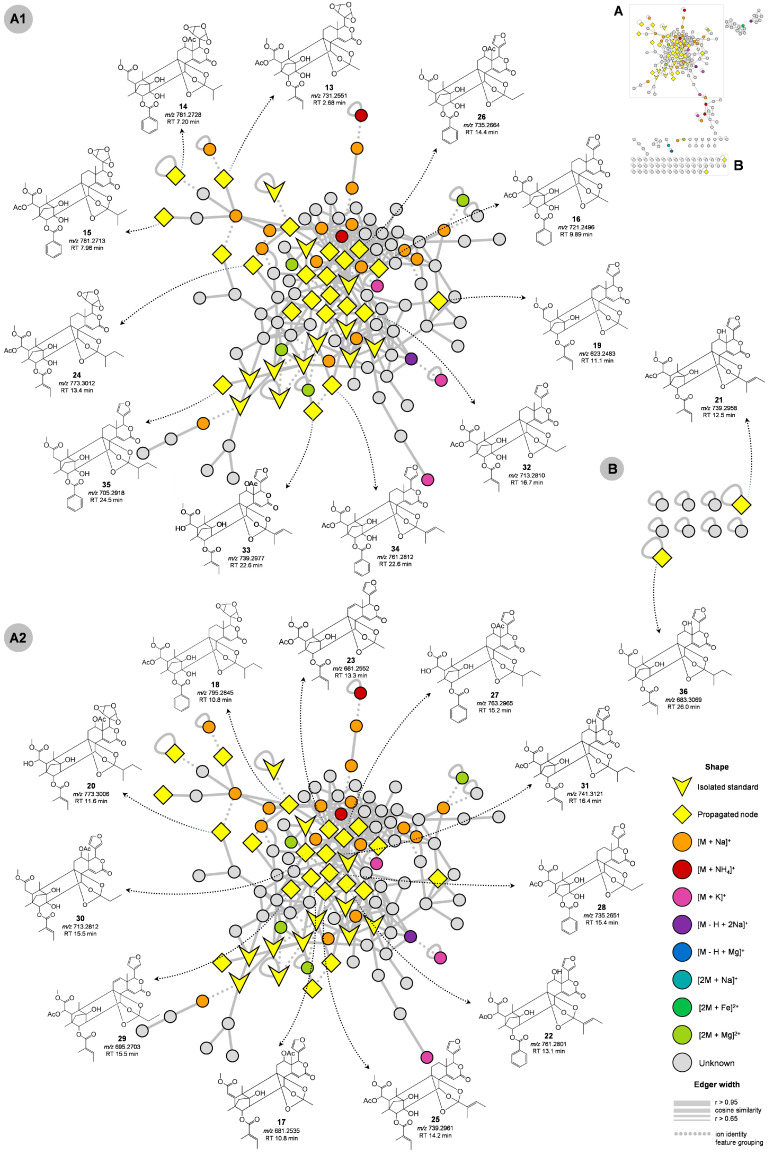
Molecular network of the main cluster (**A**) (subfigures **A1** and **A2**) and nodes group (**B**). Arrows indicate the propagated limonoids (diamond node) and their possible structure. The text below the structures indicates the parent ion and RT of the propagated compounds. The thickness limit (the line that connects two nodes) represents a spectral similarity (thicker, more similar MS Spectrum 0.65 < r > 0.95) and the dotted edge shows ion identity feature grouping.

## Data Availability

All supporting data used in this study are available from the authors.

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
