# Peer review of "Putative Identification of New Phragmaline-Type Limonoids from the Leaves of Swietenia macrophylla King: A Case Study Using Mass Spectrometry-Based Molecular Networking"

_molecules, 2023, doi:10.3390/molecules28227603_

Round 1

Reviewer 1 Report

Manuscript ID: molecules-2599863

Reviewer comments 

This research reports the putative identification of 24 phragmalin-type limonoids, described for the first time in Swietenia 2 macrophylla, making use of UPLC-MS/MS analysis and Global Natural Products Social Molecular Networking platform. The authors also propose valuable structural information of the tentatively identified compounds, explaining the fragmentation patterns from tandem MS data. The group has strong experience in natural products identification, since they performed isolation and dereplication of the 12 limonoids used as standards. In general, the work is reasonably well presented and organized and the manuscript fits into the scope of Molecules journal. The methodological approach is appropriate and the results obtained in this work might be useful in the discovery of new drugs from natural products research. 

In my opinion the manuscript can be accepted for publication in Molecules after moderate revision.

Some suggestions are the following:

-       I miss a general scheme showing a workflow to identify limonoid-type structures, based on typical product ions and neutral losses observed in this work. This figure should clarify the discussion section.

-       Lines 233-235: “…all phragmalin-type limonoids which were annotated in this study have a functional group linked to the carbon skeleton…”, “…benzoate, , and orthoester”

This sentence seems to be quite trivial, please rewrite and correct typographical errors in this sentence.

-       Line 237: Via (1). Please indicate which is via (2)

-       Lines 255-257: Please correct typographical errors (missing parenthesis) and grammatical errors in this sentence.

-       Line 276. Number starting a sentence, please correct.

-       Lines 281-283: Please correct the grammatical error in this sentece.

-       Line 291: “…crushed material was solubilized with 500 mL of…” Solubilization is not the appropriate terminology here. Please correct.

-       Line 300: Number starting a sentence, please correct.

Please correct minor typographical and grammatical errors all though the manuscript

Author Response

Reviewer 1

General comments:

This research reports the putative identification of 24 phragmalin-type limonoids, described for the first time in Swietenia macrophylla, making use of UPLC-MS/MS analysis and Global Natural Products Social Molecular Networking platform. The authors also propose valuable structural information of the tentatively identified compounds, explaining the fragmentation patterns from tandem MS data. The group has strong experience in natural products identification, since they performed isolation and dereplication of the 12 limonoids used as standards. In general, the work is reasonably well presented and organized and the manuscript fits into the scope of Molecules journal. The methodological approach is appropriate and the results obtained in this work might be useful in the discovery of new drugs from natural products research.

In my opinion the manuscript can be accepted for publication in Molecules after moderate revision.

Some suggestions are the following:

Response: Thank you for your time and effort invested in reviewing this work. We have made changes based upon your valuable input.

Specific comments:

Comment 1:

I miss a general scheme showing a workflow to identify limonoid-type structures, based on typical product ions and neutral losses observed in this work. This figure should clarify the discussion section.

Response: This is an excellent comment. An overview figure showing the workflow used in this study was added at the end of the introduction section (page 2, lines 76-77).

Comment 2:

Lines 233-235: “…all phragmalin-type limonoids which were annotated in this study have a functional group linked to the carbon skeleton…”, “…benzoate, , and orthoester”

Response: Sorry for the lack of clarity. We have revised the text as “Overall, all phragmalin-type limonoids which were annotated in this study have a functional group such as hydroxyl, acetate, furan, tiglate, benzoate, and orthoester linked to the carbon skeleton illustrated in Figure 1” (page 9, lines 240-242).

Comment 3:

This sentence seems to be quite trivial, please rewrite and correct typographical errors in this sentence.

Line 237: Via (1). Please indicate which is via (2)

Response: Thank you. It was addressed in the manuscript. We have now rewritten the sentence as “(2) by cleavage at 8,9,30-orthoester unit” (page 9, lines 244-245).

Comment 4:

Lines 255-257: Please correct typographical errors (missing parenthesis) and grammatical errors in this sentence.

Response: Sorry for the lack of clarity. It was addressed in the manuscript. We have now rewritten, the text as follows:

“Notably, in this study, four phragmalin orthoester limonoids (21, 22, 31, and 36) are described, potentially for the first time. Furthermore, eleven other limonoids (17, 19-23, 27, 29, 31, 33, and 36) did not exhibit hydroxyl or acetate groups at C-2, which is a common modification recently reported in an isolated limonoid from the roots of S. macrophylla”. (page 9, lines 262-265).

Comment 5:

Line 276. Number starting a sentence, please correct.

Response: Sorry for the lack of clarity. It was addressed in the manuscript. We have now rewritten the sentence as “Twelve isolated limonoids from S. macrophylla leaves were used as standards (identification at level 1).” (page 10, lines 282-283).

Comment 6:

Lines 281-283: Please correct the grammatical error in this sentence.

Response: Thank you for caching at this point. It was rewritten in the revised text as “Dr. Orlando Shigueo Ohashi identified the species, which was then deposited in the Federal Rural University of the Amazon's Herbarium (voucher specimen: 1330a). The National System for Managing Genetic Heritage and Associated Traditional Knowledge (SISGEN) issued authorization to access the Brazilian genetic heritage with the permission code A678D8C.” (page 10, lines 287-289).

Comment 7:

 Line 291: “…crushed material was solubilized with 500 mL of…” Solubilization is not the appropriate terminology here. Please correct.

Response:

Thank you. We have now rewritten the sentence as “After that, a total of 25 g of crushed material was extracted with 500 mL of dichloromethane”. (page 10, lines 297).

Comment 8:

Line 300: Number starting a sentence, please correct.

Response:

Sorry again for starting a sentence with a number. We have revised the sentence as “A total of 10 mg of dried extract suspended in 1 mL of H2O/CH3CN (20:80, v/v) was filtered in the C18 cartridge yielding 3.2 mg.” (page 10, line 307).

Comment 9:

Comments on the Quality of English Language:

Please correct minor typographical and grammatical errors all though the manuscript.

Response: Sorry for the lack of text quality. We have now rewritten and improved several sentences in the revised manuscript.

Reviewer 2 Report

The manuscript entitled «New phragmaline-type limonoids from the leaves of Swietenia  macrophylla King: An annotation using the molecular network approach and mass spectrometry» report a powerful analytical strategy  to study plant extract bioactive content by the use of  mass spectrometry-based molecular networking leading to reveal new putative phragmalin-type limonoids  from their mass spectrometry signatures. Thanks  to the in-house MS database  constituted from isolated and unambigously identified phragmaling-limonoids compounds from Switenia macrophylla,  the authors accomplished a highly smart MS fragmentometry network allowing annotation of new putative compounds within a high level confidence. Presented with a well-documented literature data compilation, the paper gathered detailed fragmentation processes of limonoid compounds  well illustrated by schemes and  clear figures of molecular networking clusters. The manuscript is well written within and well conducted discussion but I would like just to address to the authors one question by asking them to add the limits of the presented powerful method to fulfill a structural elucidation of new natural compounds from a complex mixture such as the studied  plant extract and to present some perspectives issues.

So, I recommend the acceptation of this paper to be published in Molecules journal after this required minor revision.

Author Response

Reviewer 2

Comment:

The manuscript entitled “New phragmaline-type limonoids from the leaves of Swietenia macrophylla King: An annotation using the molecular network approach and mass spectrometry” report a powerful analytical strategy to study plant extract bioactive content by the use of mass spectrometry-based molecular networking leading to reveal new putative phragmalin-type limonoids from their mass spectrometry signatures. Thanks to the in-house MS database constituted from isolated and unambigously identified phragmaling-limonoids compounds from Switenia macrophylla, the authors accomplished a highly smart MS fragmentometry network allowing annotation of new putative compounds within a high level confidence. Presented with a well-documented literature data compilation, the paper gathered detailed fragmentation processes of limonoid compounds well illustrated by schemes and clear figures of molecular networking clusters. The manuscript is well written within and well conducted discussion but I would like just to address to the authors one question by asking them to add the limits of the presented powerful method to fulfill a structural elucidation of new natural compounds from a complex mixture such as the studied plant extract and to present some perspectives issues.

So, I recommend the acceptation of this paper to be published in Molecules journal after this required minor revision.

Response: Thank you for your time and effort invested in reviewing this manuscript. We are encouraged by the generous comments about the quality of our work. We have now added a sentence describing the limitation at the end of the conclusion as follows:

“(…) However, it is essential to acknowledge certain limitations. While our aim is consistent and rapid chemical characterization using tandem mass spectrometry data, we can only assign putative annotations, which represent the planar structure. Therefore, considering these limitations, we emphasize the necessity of NMR experiments on the compounds annotated here. Such experiments remain crucial for identifying the three-dimensional structure of these molecules” (page 12, lines 381-387).

Reviewer 3 Report

The manuscript is well-prepared and perfectly drawn and written.

There are 44 responses to the question on Swietenia macrophylla - limonoids in WoS. Most of the papers, that describe the concrete compounds give data on their chirality. This manuscript gives on the chirality of compounds just a partial information (even on standards !!). This is in natural products chemistry a failure.

As a natural product chemist, I object to the characterization of compounds just on the LC/MS characteristic. Moreover, the  TIC and MS do not match in presented pictures. Partial help may be a mixed chromatogram, e.g. Analysis of a complex mixture of at least 12 compounds seems to me as too speculative.

So, despite the perfect presentation no data are given on the structures of compounds isolated, just MS, that is not enough.

Overall this well-prepared manuscript shall be considered just as a speculative estimation, supported by sophisticated data manipulation.

So, for me, this manuscript is not publishable, unless the authors give more information about structure characterization. In the first attempt, I may waive for the lack of stereochemical description. 

Author Response

Reviewer 3

Comment 1:

The manuscript is well-prepared and perfectly drawn and written.

Response: Thank you for your time and effort invested in reviewing this manuscript. We are encouraged by the generous comments about the quality of our work.

Comment 2:

There are 44 responses to the question on Swietenia macrophylla - limonoids in WoS. Most of the papers, that describe the concrete compounds give data on their chirality. This manuscript gives on the chirality of compounds just a partial information (even on standards !!). This is in natural products chemistry a failure.

Response: Sorry for the lack of chirality of the internal standards. We have added the Chirality of all 12 ISTD incorporated in this manuscript based on studies published by our group (Please, see Figure 4, page 7).

Comment 3:

As a natural product chemist, I object to the characterization of compounds just on the LC/MS characteristic. Moreover, the TIC and MS do not match in presented pictures. Partial help may be a mixed chromatogram, e.g. Analysis of a complex mixture of at least 12 compounds seems to me as too speculative.

Response: We totally understand your concerns about metabolite characterization by MS only. We would like to highlight that we are not attributing the total structure for those 24 new limonoids since it is a putative annotation (level 3) based on international assignment levels (the reference was cited in the text, page 4, line 98). We have now added a sentence encouraging that NMR experiments to any confirmation of those putative structures as “Table S1 (supplementary information) shows standards (1-12) and the putatively annotated compounds (13-36), and most of them are described for the first time in the leaves of the S. macrophylla. We highlight that for those putatively annotated compounds, isolation and NMR experiments are still needed and crucial for advancing our understanding of the three-dimensional structure of limonoids.” (page 3, lines 83-88).

Regarding the RT shift between the TIC of the extract against the TIC containing 12 limonoids, it is due the matrix effects, and we are highlighting it in the Figure 2. Also, all ISTD and their MS/MS spectra were matched against the MS/MS spectra collected for them in the extract.

Comment 4:

So, despite the perfect presentation no data are given on the structures of compounds isolated, just MS, that is not enough.

Response: We totally understand your concerns about metabolite characterization by MS only. We would like to highlight that we are not attributing the total structure for those 24 new limonoids since it is a putative annotation (level 3) based on international assignment levels (the reference was cited in the text, page 4, line 98). We have now added a sentence encouraging that NMR experiments to any confirmation of those putative structures as “Table S1 (supplementary information) shows standards (1-12) and the putatively annotated compounds (13-36), and most of them are described for the first time in the leaves of the S. macrophylla. We highlight that for those putatively annotated compounds, isolation and NMR experiments are still needed and crucial for advancing our understanding of the three-dimensional structure of limonoids.” (page 3, lines 83-88). Also, “(…) However, it is essential to acknowledge certain limitations. While our aim is consistent and rapid chemical characterization using tandem mass spectrometry data, we can only assign putative annotations, which represent the planar structure. Therefore, considering these limitations, we emphasize the necessity of NMR experiments on the compounds annotated here. Such experiments remain crucial for identifying the three-dimensional structure of these molecules” (page 12, lines 381-387).

Comment 5:

Overall this well-prepared manuscript shall be considered just as a speculative estimation, supported by sophisticated data manipulation. So, for me, this manuscript is not publishable, unless the authors give more information about structure characterization. I may waive for the lack of stereochemical description in the first attempt.

Response: We totally understand your concerns about metabolite characterization by MS only. We have now added a sentence describing the limitation at the end of the conclusion as follows: “(…) However, it is essential to acknowledge certain limitations. While our aim is consistent and rapid chemical characterization using tandem mass spectrometry data, we can only assign putative annotations, which represent the planar structure. Therefore, considering these limitations, we emphasize the necessity of NMR experiments on the compounds annotated here. Such experiments remain crucial for identifying the three-dimensional structure of these molecules” (page 12, lines 381-387).

Round 2

Reviewer 3 Report

The authors significantly improved the manuscript. So, I ca now (with still some hesitations) agree with accepting the paper.